# Drug-induced vitiligo: a real-world pharmacovigilance analysis of the FAERS database

Nonger Shen[1], Qingxia Fang[1], Yue Wu[1], Lan Lan[1], Fangfang Ma[2]*

1 Center for Clinical Pharmacy, Cancer Center, Department of Pharmacy, Zhejiang Provincial People's Hospital (Affiliated People's Hospital), Hangzhou Medical College, Hangzhou, Zhejiang, China, 2 Geriatric Medicine Center, Department of Nursing, Zhejiang Provincial People's Hospital (Affiliated People's Hospital), Hangzhou Medical College, Hangzhou, Zhejiang, China

* 391140197@qq.com

## Abstract

### Background

In recent years, with the expanding use of novel therapeutics such as immune check-point inhibitors and monoclonal antibodies, reports of drug-induced vitiligo have been increasing. This study aimed to identify drugs associated with vitiligo using the FDA Adverse Event Reporting System (FAERS).

### Methods

A retrospective disproportionality analysis was performed on FAERS reports from the first quarter of 2004 to the fourth quarter of 2024. Disproportionality signals were assessed using the Reporting odds ratio (ROR) and Bayesian confidence propagation neural network (BCPNN).

### Results

The present study identified 1,910 cases of vitiligo, with a median age of 54 years (interquartile range [IQR]: 40.0–66.0). The gender distribution among these cases was 45.9% female, 35.9% male, and 18.3% with undetermined gender. The three most frequently reported drugs were nivolumab (152 cases), pembrolizumab (132 cases), and dupilumab (77 cases). Forty-six drugs showed significant positive vitiligo signals. The three strongest signals (based on ROR) were mogamulizumab (ROR 73.93, 95% confidence interval [CI] 39.62–137.94; the lower 95% CI for the information component [$IC_{025}$] 2.40), imiquimod (ROR 72.00, 95% CI 43.24–119.89; $IC_{025}$ 3.00) and chloroquine (ROR 53.33, 95% CI 17.13–166.02; $IC_{025}$ 0.47). Notably, 80.4% (37/46) of these drugs lacked vitiligo warnings in their FDA-approved labels.

**Data availability statement:** The datasets utilized in this study are publicly available from the FAERS database online repository: https://fis.fda.gov/extensions/FPD-QDE-FAERS/FPD-QDE-FAERS.html. This analysis encompassed data from Q1 2004 to Q4 2024.

**Funding:** The author(s) received no specific funding for this work.

**Competing interests:** The authors have declared that no competing interests exist.

## Conclusion

This study has uncovered an extensive catalog of drugs with the potential to induce vitiligo. While these findings are based on pharmacovigilance signals and require further validation, patients receiving these medications should be closely monitored for the development of vitiligo.

## 1. Introduction

Vitiligo is an acquired depigmenting disorder characterized by the presence of well-demarcated white patches of varying sizes on the skin [1]. The distribution of these patches may be localized to a specific body area or generalized throughout the body, particularly on exposed areas such as the face, hands, and feet. The etiology of vitiligo has yet to be fully elucidated. However, it is considered a multifactorial disease involving genetic, environmental, and immunological factors [2,3]. The prevalence of this condition ranges from 0.5% to 2% among the global population [4]. It occurs uniformly across all genders, age groups, and skin types [5]. While vitiligo is not lethal and is not contagious, its impact on a patient's appearance can result in a considerable psychosocial burden, potentially precipitating psychological issues such as depression and anxiety [6–8].

In recent years, as novel therapeutics, including immune checkpoint inhibitors (ICIs), monoclonal antibodies, and immunomodulators, have increasingly been utilized in clinical practice, there has been an increase in reports of drug-induced vitiligo [9]. Studies indicated that approximately 10% to 28% of melanoma patients treated with ICIs may develop vitiligo [10–12]. Furthermore, 5% of patients undergoing talimogene laherparepvec therapy have presented with vitiligo [13]. However, the clinical understanding of drug-induced vitiligo remains limited, as knowledge regarding it mainly stems from sporadic case reports.

Pharmacovigilance studies play a crucial role in identifying previously unreported adverse drug reactions (ADR) during post-marketing surveillance. Despite the exploration of associations between drugs and vitiligo in previous studies, existing research is largely confined to specific drug classes. Yang et al. [14] utilized the Food and Drug Administration (FDA) Adverse Event Reporting System (FAERS) database to investigate the association between vitiligo risk and antineoplastic agents, while Lu et al. [15] examined type 2 immune inhibitors. Furthermore, a 2020 case/non-case study using VigiBase only reported eight potential drug associations [16]. To date, a comprehensive analysis covering all pharmaceuticals has not yet been conducted specifically for drug-induced vitiligo. This knowledge gap is especially concerning given the accelerating approval of novel immunotherapies and the increasing number of case reports of drug-induced vitiligo.

The FAERS database, a global platform for spontaneous adverse event reporting, offers a vast and diverse dataset with standardized information. Its large sample size strengthens the reliability of disproportionality analysis in cases of rare events, such as drug-induced vitiligo [17,18]. The objective of this retrospective study is to employ

the FAERS database to conduct a screening of potential drugs associated with vitiligo across all therapeutic categories and to identify agents that lack appropriate safety alerts in their prescribing information. The findings are expected to enhance our understanding of this complex disease, provide references for clinical practice, and contribute to the development of more effective strategies for the prevention and management of drug-induced vitiligo.

## 2. Methods

### 2.1. Data source

The FAERS data files, ranging from the first quarter of 2004 to the fourth quarter of 2024, were obtained from the FDA website (https://fis.fda.gov/extensions/FPD-QDE-FAERS/FPD-QDE-FAERS.html). The FAERS structure comprises seven core datasets: demographic/administrative records (DEMO), drug exposure details (DRUG), adverse reaction descriptions coded using MedDRA (REAC), patient outcomes (OUTC), reporting sources (RPSR), therapy duration (THER), and drug indications (INDI).

### 2.2. Inclusion and exclusion criteria

To be included in this study, reports had to fulfill three criteria: first, they must document an adverse event coded as "vitiligo" using the Preferred Term (PT) in MedDRA version 27.1 (code: 10047642) within the REAC dataset; second, the associated drug in the DRUG dataset must be classified as a "Primary Suspect (PS)"; and third, they must remain after the implementation of the deduplication protocol.

Reports were excluded if they belonged to any of the following categories: duplicate entries, where for records with identical CASEIDs, only the one with the most recent FDA receipt date (FDA_DT) was retained, and in cases of matching CASEID and FDA_DT, the entry with the lower PRIMARYID (unique report identifier) was excluded; reports where the drug was categorized as "Secondary Suspect (SS)", "Concomitant (C)", or "Interacting (I)" in the DRUG dataset; or reports lacking clear documentation of vitiligo.

The standardization of drug nomenclature was implemented to address synonymic variations by leveraging two authoritative resources: DrugBank (https://go.drugbank.com/) and the Medical Subject Headings (MeSH) thesaurus (https://www.ncbi.nlm.nih.gov/mesh/). Each drug was subsequently classified according to the Anatomical Therapeutic Chemical (ATC) classification system, maintained by the WHO Collaborating Centre for Drug Statistics Methodology, to facilitate organ-system and mechanistic analyses.

### 2.3. Statistical analysis

Descriptive analysis was employed to delineate the demographic and clinical features of vitiligo-associated AE reports. Variables of interest included patient age, sex, body weight, geographic origin, reporting year, reporters, and clinical outcomes. Disproportionality analyses were conducted using two complementary algorithms: the frequentist Reporting odds ratio (ROR) and the Bayesian confidence propagation neural network (BCPNN) [19]. The ROR is a frequentist statistical method known for its high sensitivity in detecting initial signals. In contrast, BCPNN is a Bayesian approach that effectively mitigates false positives associated with low-frequency events through shrinkage induced by Bayesian priors, thereby enhancing the robustness of the analysis [20,21]. The formulae for each method are shown in S1 Table. For the ROR method, a signal was considered statistically significant if the lower bound of the 95% confidence interval (CI) exceeded 1, accompanied by a minimum of three reported cases. For the BCPNN framework, disproportionality was quantified using Information Components (IC), with a positive signal defined by an IC lower bound ($IC_{025}$) > 0. To enhance the specificity of signal detection and mitigate methodological biases, only drug-vitiligo associations satisfying both criteria were classified as robust pharmacovigilance signals in this study.

All statistical computations and visualizations were performed using R software version 4.4.2 and Microsoft Excel 2019.

## 3. Results

### 3.1. Descriptive analysis

A total of 22,249,476 AE reports were extracted from the FAERS database between Q1 2004 and Q4 2024 (Fig 1). After de-duplication and data cleaning, 1,910 cases were found to be associated with vitiligo. The demographic characteristics of the cases included in this study are presented in Table 1, which shows that males and females constituted 35.9% and 45.9% of the cases, respectively. The median age of the patients in the reported cases was 54.0 (interquartile range [IQR]: 40.0–66.0). The majority of cases occurred in patients aged 50–60 years (237 cases, 12.4%), followed by those aged 60–70 years (223 cases, 11.7%) and aged 40–50 years (197 cases, 10.3%) (Fig 2). The annual number of reported cases of vitiligo exhibited a fluctuating growth trend, peaking at 315 cases in 2024. These cases were mainly reported by physicians (34.9%), consumers (30.1%), and other health professionals (28.6%). Most reports originated from the United States (51.5%), followed by France (7.3%), Germany (4.4%), the United Kingdom (4.4%), and Canada (3.2%). Nivolumab was the most frequently reported drug (152 cases), followed by pembrolizumab (132 cases) and dupilumab (77 cases) (Fig 3).

### 3.2. Disproportionality analysis

A total of 46 drugs were identified as potentially associated with vitiligo (Fig 4). The top 3 drugs with significant positive signals based on the ROR were mogamulizumab (ROR 73.93, 95%CI 39.62–137.94; $IC_{025}$ 2.40), imiquimod (ROR 72.00, 95% CI 43.24–119.89; $IC_{025}$ 3.00), and chloroquine (ROR 53.33, 95% CI 17.13–166.02; $IC_{025}$ 0.47). Among these 46 drugs with positive signals, 29 (63.0%) belonged to the therapeutic category of antineoplastic agents and immunomodulating agents (Table 2), including programmed cell death protein 1/programmed death-ligand 1 (PD-1/PD-L1) inhibitors, cyclin-dependent kinase 4/6 (CDK4/6) inhibitors, B-Raf proto-oncogene (BRAF) inhibitors, mitogen-activated protein kinase kinase (MEK) inhibitors, cytotoxic T-lymphocyte-associated protein 4 (CTLA-4) inhibitors, alkylating agents, and aromatase inhibitors. Notably, vitiligo was not listed as an adverse effect in the package inserts of 37 (80.4%) of these drugs.

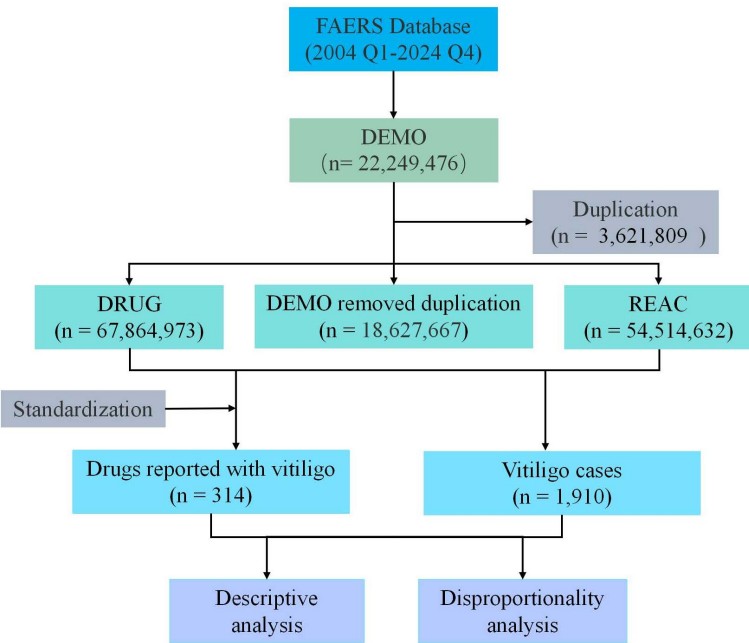

**Fig 1. Flow chart for the identification of vitiligo cases from the US Food and Drug Administration Adverse Event System.**

**Table 1. Characteristics of drug-associated vitiligo in FAERS.**

| Characteristics | Values |
|---|---|
| Number of total cases | 1,910 |
| **Gender** | |
| Female | 876 (45.9%) |
| Male | 685 (35.9%) |
| Unknown | 349 (18.3%) |
| **Age (years)** | |
| Mean ± SD | 51.4 ± 19.3 |
| Median (IQR) | 54.0 (40.0-66.0) |
| Unknown | 791 (41.4%) |
| **Outcomes** | |
| Death | 43 (2.3%) |
| Life-Threatening | 29 (1.5%) |
| Hospitalization | 254 (13.3%) |
| Disability | 66 (3.5%) |
| Congenital Anomaly | 3 (0.2%) |
| Required Intervention | 4 (0.2%) |
| Other Serious | 814 (42.6%) |
| Unknown | 697 (36.5%) |
| **Reporters** | |
| Physician | 667 (34.9%) |
| Pharmacist | 63 (3.3%) |
| Other health-professional | 547 (28.6%) |
| Consumer | 575 (30.1%) |
| Lawer | 17 (0.9%) |
| Unknown | 41 (2.1%) |
| **TOP 5 Report Countries** | |
| United States | 983 (51.5%) |
| France | 140 (7.3%) |
| Germany | 84 (4.4%) |
| United Kingdom | 84 (4.4%) |
| Canada | 62 (3.2%) |

IQR, interquartile range; SD, standard deviation.

## 4. Discussion

The present study identified and classified the drugs most closely associated with the induction of vitiligo in one of the world's largest adverse event reporting databases, with data covering the period from January 2004 to December 2024. The analysis demonstrated a fluctuating annual increase in the number of reported cases of drug-induced vitiligo. The top five drugs determined according to the number of case reports in the dataset were nivolumab, pembrolizumab, dupilumab, ipilimumab, and secukinumab, all of which were monoclonal antibody biologic agents. This observation suggests a potential correlation between the increasing number of drug-induced vitiligo cases and the growing clinical application of novel biologic agents and targeted therapies.

Our study revealed a higher prevalence of reported drug-induced vitiligo cases in female patients, which aligns with previous epidemiological data on non-drug-induced vitiligo [22–25]. In the United States, analysis of vitiligo cases

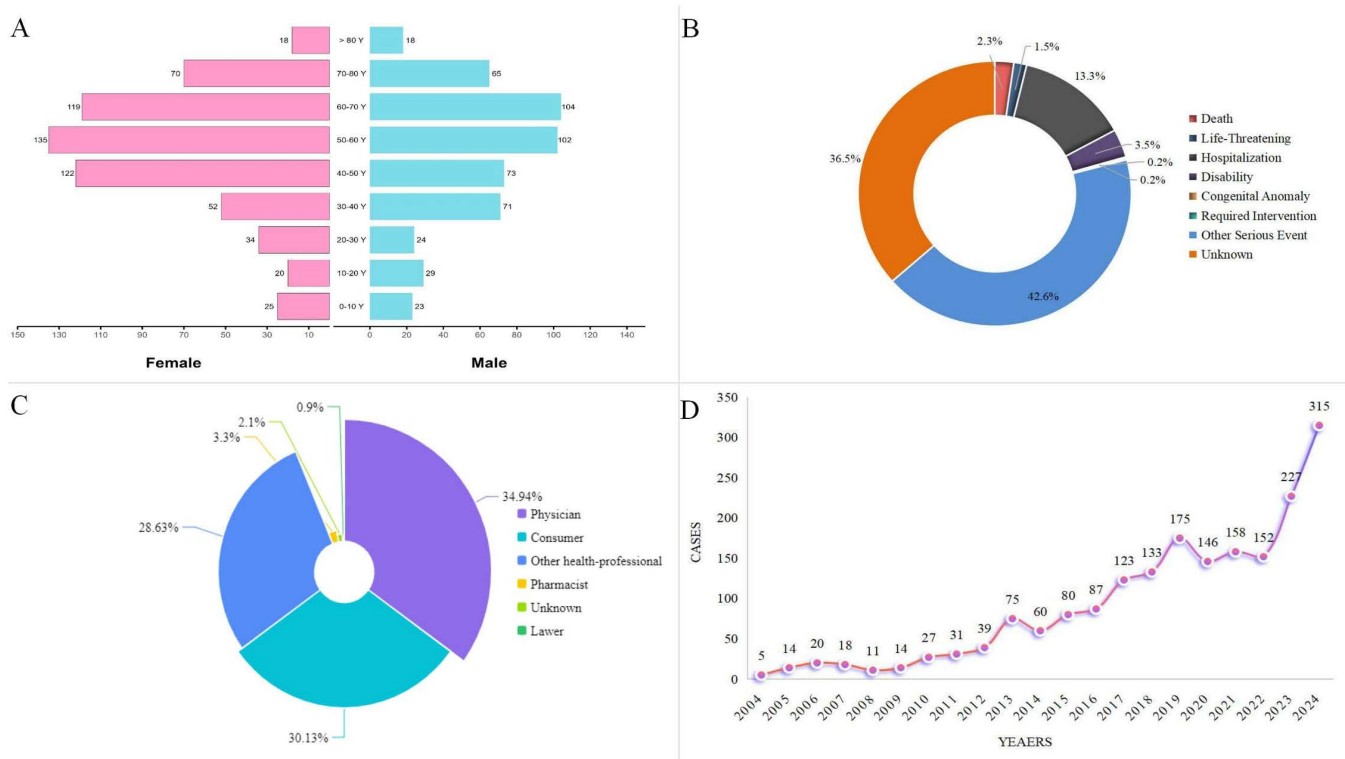

**Fig 2. Characteristics of Vitiligo Reports from the FDA Adverse Event Reporting System Database. (A)** Gender and Age Distribution of Subjects: Pink bars represent female subjects, and teal bars represent male subjects, with age groups on the x-axis and count of reports on the y-axis. **(B)** Outcome Distribution: Pie chart showing proportions of different adverse event outcomes based on report data. **(C)** Reporter Distribution: Pie chart categorizing report sources and their respective percentages. **(D)** Annual Reporting Frequencies: Line graph depicting the number of vitiligo-related reports (y-axis) over years (x-axis).

| Drugs | Number of cases | | Female n (%) | Male n (%) | Age median(IQR) |
|---|---|---|---|---|---|
| Nivolumab | | 152 | 52 (34.2%) | 86 (56.6%) | 64.0 (52.0-72.0) |
| Pembrolizumab | | 132 | 63 (47.7%) | 60 (45.5%) | 64.0 (53.0-72.0) |
| Dupilumab | | 77 | 31 (40.3%) | 37 (48.1%) | 34.0 (12.0-55.3) |
| Adalimumab | | 74 | 33 (44.6%) | 31 (41.9%) | 43.5 (35.8-53.0) |
| Ipilimumab | | 67 | 20 (29.9%) | 40 (59.7%) | 58.0 (48.0-69.0) |
| Secukinumab | | 49 | 26 (53.1%) | 19 (38.8%) | 53.0 (44.0-53.0) |
| Infliximab | | 45 | 14 (31.1%) | 13 (28.9%) | 46.0 (21.0-56.0) |
| Etanercept | | 40 | 28 (70.0%) | 10 (25.0%) | 55.5 (45.3-63.5) |
| Ribociclib | | 36 | 33 (91.7%) | 0 (0%) | 58.0 (55.0-60.5) |
| Ustekinumab | | 31 | 10 (32.3%) | 12 (38.7%) | 52.0 (45.0-58.0) |

**Fig 3. Top 10 Drugs most frequently associated with Vitiligo.** IQR, interquartile range.

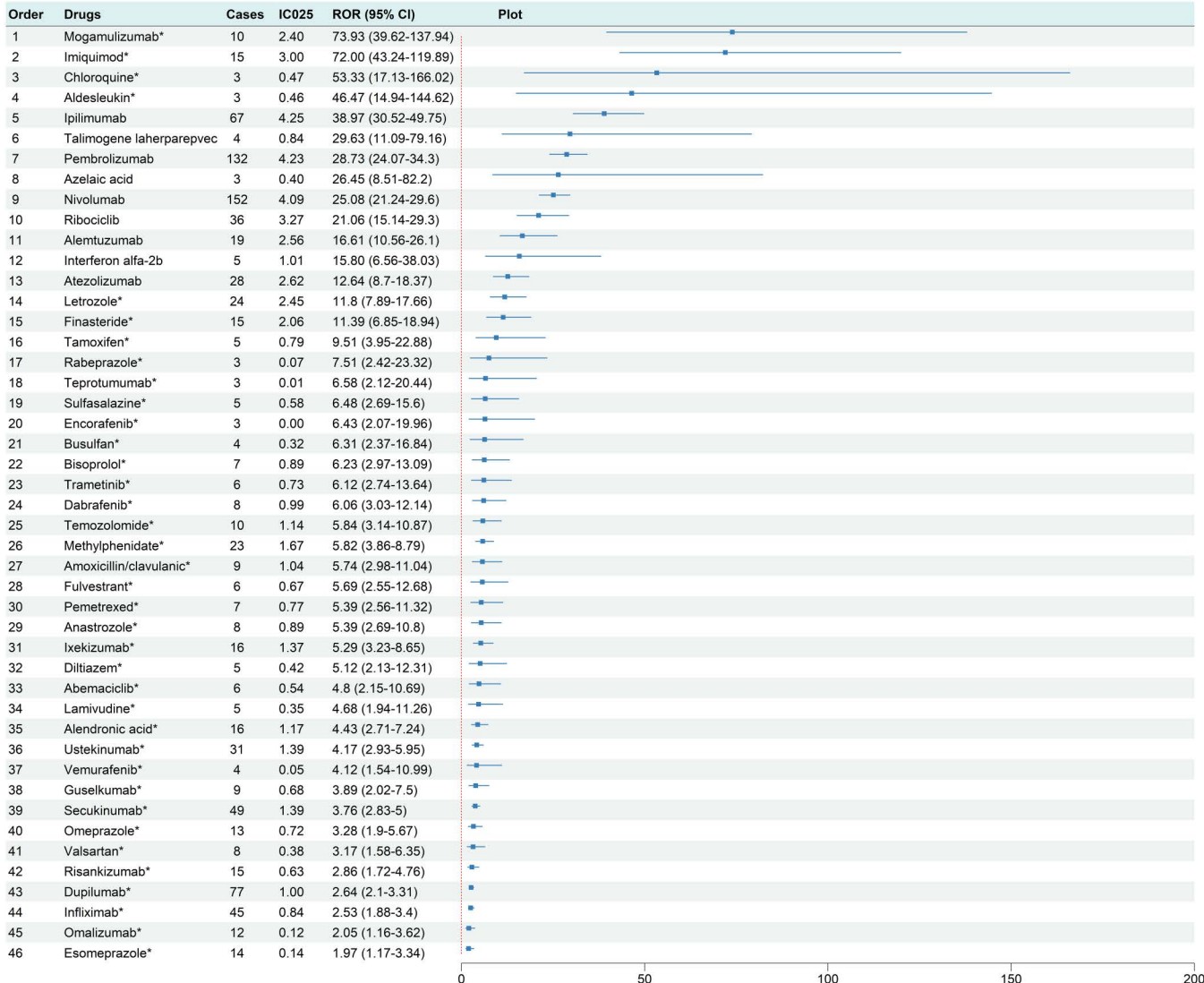

| Order | Drugs | Cases | IC025 | ROR (95% CI) |
|---|---|---|---|---|
| 1 | Mogamulizumab* | 10 | 2.40 | 73.93 (39.62-137.94) |
| 2 | Imiquimod* | 15 | 3.00 | 72.00 (43.24-119.89) |
| 3 | Chloroquine* | 3 | 0.47 | 53.33 (17.13-166.02) |
| 4 | Aldesleukin* | 3 | 0.46 | 46.47 (14.94-144.62) |
| 5 | Ipilimumab | 67 | 4.25 | 38.97 (30.52-49.75) |
| 6 | Talimogene laherparepvec | 4 | 0.84 | 29.63 (11.09-79.16) |
| 7 | Pembrolizumab | 132 | 4.23 | 28.73 (24.07-34.3) |
| 8 | Azelaic acid | 3 | 0.40 | 26.45 (8.51-82.2) |
| 9 | Nivolumab | 152 | 4.09 | 25.08 (21.24-29.6) |
| 10 | Ribociclib | 36 | 3.27 | 21.06 (15.14-29.3) |
| 11 | Alemtuzumab | 19 | 2.56 | 16.61 (10.56-26.1) |
| 12 | Interferon alfa-2b | 5 | 1.01 | 15.80 (6.56-38.03) |
| 13 | Atezolizumab | 28 | 2.62 | 12.64 (8.7-18.37) |
| 14 | Letrozole* | 24 | 2.45 | 11.8 (7.89-17.66) |
| 15 | Finasteride* | 15 | 2.06 | 11.39 (6.85-18.94) |
| 16 | Tamoxifen* | 5 | 0.79 | 9.51 (3.95-22.88) |
| 17 | Rabeprazole* | 3 | 0.07 | 7.51 (2.42-23.32) |
| 18 | Teprotumumab* | 3 | 0.01 | 6.58 (2.12-20.44) |
| 19 | Sulfasalazine* | 5 | 0.58 | 6.48 (2.69-15.6) |
| 20 | Encorafenib* | 3 | 0.00 | 6.43 (2.07-19.96) |
| 21 | Busulfan* | 4 | 0.32 | 6.31 (2.37-16.84) |
| 22 | Bisoprolol* | 7 | 0.89 | 6.23 (2.97-13.09) |
| 23 | Trametinib* | 6 | 0.73 | 6.12 (2.74-13.64) |
| 24 | Dabrafenib* | 8 | 0.99 | 6.06 (3.03-12.14) |
| 25 | Temozolomide* | 10 | 1.14 | 5.84 (3.14-10.87) |
| 26 | Methylphenidate* | 23 | 1.67 | 5.82 (3.86-8.79) |
| 27 | Amoxicillin/clavulanic* | 9 | 1.04 | 5.74 (2.98-11.04) |
| 28 | Fulvestrant* | 6 | 0.67 | 5.69 (2.55-12.68) |
| 30 | Pemetrexed* | 7 | 0.77 | 5.39 (2.56-11.32) |
| 29 | Anastrozole* | 8 | 0.89 | 5.39 (2.69-10.8) |
| 31 | Ixekizumab* | 16 | 1.37 | 5.29 (3.23-8.65) |
| 32 | Diltiazem* | 5 | 0.42 | 5.12 (2.13-12.31) |
| 33 | Abemaciclib* | 6 | 0.54 | 4.8 (2.15-10.69) |
| 34 | Lamivudine* | 5 | 0.35 | 4.68 (1.94-11.26) |
| 35 | Alendronic acid* | 16 | 1.17 | 4.43 (2.71-7.24) |
| 36 | Ustekinumab* | 31 | 1.39 | 4.17 (2.93-5.95) |
| 37 | Vemurafenib* | 4 | 0.05 | 4.12 (1.54-10.99) |
| 38 | Guselkumab* | 9 | 0.68 | 3.89 (2.02-7.5) |
| 39 | Secukinumab* | 49 | 1.39 | 3.76 (2.83-5) |
| 40 | Omeprazole* | 13 | 0.72 | 3.28 (1.9-5.67) |
| 41 | Valsartan* | 8 | 0.38 | 3.17 (1.58-6.35) |
| 42 | Risankizumab* | 15 | 0.63 | 2.86 (1.72-4.76) |
| 43 | Dupilumab* | 77 | 1.00 | 2.64 (2.1-3.31) |
| 44 | Infliximab* | 45 | 0.84 | 2.53 (1.88-3.4) |
| 45 | Omalizumab* | 12 | 0.12 | 2.05 (1.16-3.62) |
| 46 | Esomeprazole* | 14 | 0.14 | 1.97 (1.17-3.34) |

**Fig 4. Frost Plot of ROR for drugs with positive signal.** CI, confidence interval; IC025, the lower 95% CI for the information component; ROR, reporting odds ratio. * The drug package insert does not list vitiligo as an adverse event.

demonstrated a gender distribution of 57.5% females compared to 42.5% males [24]. In Tanzania, the ratio of female to male vitiligo cases was approximately 1.8:1 [25]. A similar trend was observed in South Korea, where the female-to-male ratio of vitiligo patients was approximately 1.3:1 [23]. The relatively higher incidence of vitiligo in females may be attributed to the higher prevalence of autoimmune diseases in women and their greater concern for physical appearance, which could lead to a higher diagnosis rate of vitiligo [22].

Our disproportionality analysis identified 46 drugs across 10 ATC classes with significant vitiligo associations. It is noteworthy that only 9 (19.6%) of these agents currently list vitiligo as an anticipated AE in FDA-approved package inserts. The observed discrepancy may be attributed to inherent limitations in clinical trial design and delays in updating post-marketing drug package inserts, which may affect the monitoring, recognition, and management of drug-induced vitiligo in clinical practice. It suggested the necessity for dynamic post-marketing pharmacovigilance.

**Table 2. Classification of Drugs Causing Drug-associated Vitiligo according ATC.**

| ATC category | Number of drugs N (%) | Drugs |
|---|---|---|
| Alimentary tract and metabolism | 4 (8.7%) | Omeprazole, Esomeprazole, Rabeprazole, Sulfasalazine |
| Cardiovascular system | 3 (6.5%) | Diltiazem, Bisoprolol, Valsartan |
| Dermatologicals | 3 (6.5%) | Imiquimod, Azelaic acid, Dupilumab |
| Genito urinary system and sex hormones | 1 (2.2%) | Finasteride |
| Antiinfectives for systemic use | 2 (4.3%) | Amoxicillin/clavulanic, Lamivudine |
| Antineoplastic and immunomodulating agents | 29 (63.0%) | Busulfan, Temozolomide, Pemetrexed, Vemurafenib, Dabrafenib, Encorafenib, Trametinib, Ribociclib, Abemaciclib, Nivolumab, Pembrolizumab, Atezolizumab, Ipilimumab, Mogamulizumab, Talimogene laherparepvec, Tamoxifen, Fulvestrant, Anastrozole, Letrozole, Interferon alfa-2b, Aldesleukin, Infliximab, Ustekinumab, Secukinumab, Ixekizumab, Guselkumab, Risankizumab, Alemtuzumab, Teprotumumab |
| Musculo – skeletal system | 1 (2.2%) | Alendronic acid |
| Nervous system | 1 (2.2%) | Methylphenidate |
| Antiparasitic products, insecticides and repellents | 1 (2.2%) | Chloroquine |
| Respiratory system | 1 (2.2%) | Omalizumab |

ATC, Anatomical Therapeutic Chemical.

The ROR is a reliable indicator for quantifying the association between drugs and AEs, with higher values indicating greater clinical relevance. Our analysis revealed the strongest association between mogamulizumab and the onset of vitiligo (ROR 73.93, 95% CI 39.62–137.94). Mogamulizumab, a monoclonal antibody targeting the CC chemokine receptor type 4 (CCR4) receptor approved for relapsed/refractory mycosis fungoides and Sézary syndrome, is known for its dermatologic toxicities, including rash, Stevens-Johnson syndrome, and toxic epidermal necrolysis [26]. A recently published case report detailed the case of an 83-year-old patient diagnosed with IV-A2 mycosis fungoides who subsequently developed vitiligo after a period of six months during treatment with mogamulizumab [27]. This observation aligns with the similar latency period (6–8 months) reported in three other cases by Algarni et al. [28]. Suzuki et al. [29] hypothesized that mogamulizumab disrupts the peripheral immune checkpoint by depleting regulatory T cells (Tregs), inducing the production of autoantibodies against melanocytes. These antibodies then attack melanocytes, thus leading to vitiligo.

Imiquimod is a topical immunomodulator that activates both innate and acquired immune responses. It is widely used for the treatment of genital and perianal warts, actinic keratosis, superficial basal cell carcinoma, and other conditions [30]. In the present study, imiquimod exhibited the second strongest association signal with vitiligo. As indicated by published case reports, the onset of imiquimod-induced vitiligo occurred at an earlier stage in the treatment course of superficial basal cell carcinoma (3.5–13 weeks) than in that of condyloma acuminata (6–28 weeks) [31–33]. The depigmentation of the skin was primarily observed at or in close proximity to the application site. Prior to the manifestation of depigmentation, symptoms such as erythema, burning, pain, pruritus, and edema typically emerge. The potential mechanism of imiquimod-induced vitiligo involves interaction with Toll-like receptor 7 (TLR7) present on skin cells, which stimulates the production of various cytokines, including interferon (IFN)-α, tumor necrosis factor alpha (TNF-α), interleukin (IL)-6, IL-1, IL-8, and IL-12 [31]. These inflammatory mediators disrupt the Th1/Th2 balance, resulting in the apoptosis of melanocytes and subsequent skin depigmentation. Furthermore, imiquimod has been observed to directly inhibit melanogenesis and induce melanocyte dysfunction [34].

Chloroquine is an antimalarial agent and has also been demonstrated to possess immunomodulatory properties [35]. The findings of this study suggest that chloroquine exhibits the third strongest association signal with vitiligo, indicating

that vitiligo may be a rare ADR associated with chloroquine use. Case reports have documented the development of depigmented patches on the chest, shoulders, and forearms of a 44-year-old Hispanic female after one month of chloroquine treatment [36]. The majority of these depigmented patches persisted following the discontinuation of chloroquine for a period of two months. Furthermore, a six-year-old girl exhibited symptoms consistent with vitiligo following three weeks of chloroquine therapy [37]. The precise mechanism by which chloroquine induces vitiligo remains to be elucidated; however, it is noteworthy that all reported cases of chloroquine-induced vitiligo-like depigmentation occurred in individuals with dark or black skin [36,37].

ICIs have emerged as active therapeutic agents for the treatment of numerous cancers. Skin toxicity is a prevalent immune-related AE, and patients are frequently referred to dermatology for evaluation [24,38]. Our analysis of the FAERS database revealed that nivolumab was associated with the highest number of cases of vitiligo, followed by pembrolizumab. Meta-analysis revealed an incidence of vitiligo in patients treated with nivolumab of 7.5%, and with pembrolizumab of 8.3% [39]. The clinical utilization of ICIs predominantly encompasses the administration of CTLA-4 inhibitors, PD-1 inhibitors, and PD-L1 inhibitors. According to meta-analyses, the highest incidence of vitiligo was observed in patients receiving ICIs combination therapy (CTLA-4 combined with PD-1/PD-L1 agents, 10.1%), followed by PD-1 monotherapy (approximately 7.9%), ICIs + chemotherapy (3.7%), CTLA-4 monotherapy (3.2%), and PD-L1 monotherapy (0.54%) (P < 0.001) [40].

ICI-induced vitiligo demonstrated a significant correlation with the specific type of cancer treated, with the majority of cases manifesting in melanoma patients. Specifically, 77.5%−87.4% of cases occurred in melanoma patients, while 12.6%−23% of cases occurred in non-melanoma patients [41,42]. The prevalence of ICI-induced vitiligo was 7.9% in melanoma patients and 0.2% to 3% in other cancer groups (P < 0.001) [40]. The increased incidence of ICI-induced vitiligo in melanoma patients may be attributable to the release of shared melanocyte antigens following the destruction of melanoma tumor cells [40,41]. This results in the loss of the immune privilege of normal melanocytes induced by ICIs. ICI-induced vitiligo in non-melanoma patients may be associated with T-cell-mediated autoimmunity and may involve more extensive immune activation.

Our analysis identified interleukin inhibitors, including ustekinumab (31 cases; ROR 4.17), secukinumab (49 cases; ROR 3.76), ixekizumab (16 cases; ROR 5.29), guselkumab (9 cases; ROR 3.89), and risankizumab (15 cases; ROR 2.86), as potential triggers of vitiligo. While precise incidence rates require further quantification, accumulating case reports corroborate this association. A clinical case documented vitiligo onset in a 60-year-old patient with severe psoriatic arthritis following 4.5 months of guselkumab therapy [43]. Raman et al. [44] conducted a systematic review of studies published prior to September 17, 2022, encompassing 19 studies with 74 IL inhibitor-associated vitiligo cases. Among these, secukinumab accounted for the majority (n = 44, 59.5%), followed by ustekinumab (n = 24, 32.4%), ixekizumab (n = 5, 6.8%), and tildrakizumab (n = 1, 1%). The study revealed new-onset vitiligo in 18.9% (14/74) of cases, exacerbation of pre-existing conditions in 1.4% (1/74), while the vitiligo history remained unspecified in 60.8% (45/74) of reports. For IL-17 inhibitors (secukinumab/ixekizumab), the potential pathomechanism may involve suppression of IL-17, leading to possible disruption of the Th1/Th17 balance and enhanced release of IFN-γ, thereby exacerbating melanocyte destruction [45,46].

Finasteride, a 5-alpha-reductase inhibitor, is clinically approved for the treatment of benign prostate hyperplasia and androgenic alopecia [47]. Our analysis identified 15 cases of finasteride-associated vitiligo. Vitiligo may be a rare AE of finasteride, with currently only one case reported in the literature. Motofei et al. [48] described a case of a 52-year-old male who developed generalized vitiligo two months after discontinuing finasteride, accompanied by persistent sexual dysfunction and depression. This case has been identified as post-finasteride syndrome (PFS). PFS is a complex constellation of adverse effects that may develop and persist in patients during and/or after finasteride treatment discontinuation [49]. Dry skin, vitiligo, and leukocytoclastic vasculitis were recognized as potential dermatological manifestations of PFS [50]. The precise pathological mechanism of finasteride and vitiligo remains to be elucidated. Proposed hypotheses include finasteride-induced neuroendocrine disturbances and autoimmune disturbances [48].

 

Sporadic case reports suggested that proton pump inhibitors (PPIs) may induce depigmentation or exacerbate pre-existing vitiligo [51–53]. Schallreuter et al. [51] reported four cases of vitiligo patients who experienced disease recurrence or new skin lesions after using PPIs, and repigmentation occurred only after discontinuing esomeprazole or omeprazole. Another case described a patient in whom the surgically repigmented vitiligo lesions developed depigmentation after receiving pantoprazole treatment, and this phenomenon reversed after drug withdrawal [52]. Additionally, Kim et al. [53] reported a case of vitiligo induced by occupational exposure to PPI powders (pantoprazole and rabeprazole). Our pharmacovigilance data further confirm these observations, showing positive signals for rabeprazole (ROR 7.51, 95% CI 2.42–23.32; $IC_{025}$ 0.07), omeprazole (ROR 3.28, 95% CI 1.90–5.67; $IC_{025}$ 0.72), and esomeprazole (ROR 1.97, 95% CI 1.17–3.34; $IC_{025}$ 0.14). The mechanism by which PPIs induce vitiligo is not yet clear, but PPIs may impair melanogenesis by inhibiting the H+/K±ATPase on melanosomal membranes, disrupting the pH gradient crucial for tyrosinase activation [51]. In vitro models have shown that PPIs can reduce melanin synthesis and tyrosinase activity in melanoma cells, and zebrafish studies have confirmed dose-dependent pigment loss [54].

This study inevitably has several limitations. First, the FAERS database used in this research relies on spontaneous reports, and this passive surveillance mechanism may lead to systematic biases. Vitiligo is characterized by progressive depigmentation. Mild or local cases may not seek medical treatment in a timely manner, thus resulting in an underestimation of the true incidence. Second, due to the lack of data on the size of the drug-using population, it is impossible to calculate the incidence rate, which restricts the quantitative assessment of the risk level. Additionally, it should be noted that the use of disproportionality analysis for signal detection reveals a statistical association between drugs and AEs, rather than a causal relationship. Validation through large-scale clinical studies is required to establish a causal link. Despite these limitations, this study, through systematic analysis, has unveiled the scope of drug vigilance potentially associated with the occurrence of vitiligo. These research findings can serve as a reference for clinical medication monitoring.

## 5. Conclusion

This pharmacovigilance analysis identified 46 drugs with positive signals for drug-induced vitiligo, 80.4% of which lacked warnings in their FDA-approved labels. Antineoplastic and immunomodulating agents comprised the majority (63.0%, 29/46) of the implicated drugs, including PD-1/PD-L1 inhibitors, CTLA-4 inhibitors, CDK4/6 inhibitors, BRAF/MEK inhibitors, anti-estrogens, aromatase inhibitors, alkylating agents, interleukin inhibitors, interferons, and other monoclonal antibodies. Clinicians are advised to enhance their vigilance for vitiligo in patients undergoing these therapies. However, this study necessitates considering the inherent limitations of FAERS, along with potential confounders and biases. Thus, the analysis results should be interpreted cautiously. Future research efforts should be directed towards deepening our comprehension of the underlying associations and mechanisms that connect these drugs to vitiligo. Prospective cohort studies can provide more reliable evidence of causality and help clarify the incidence rates of drug-induced vitiligo. Mechanistic immunological investigations are necessary to elucidate the precise pathways through which these drugs induce vitiligo. Integration of real-world data can further validate these associations and guide clinical risk management.

## Supporting information

**S1 Table. Algorithms employed for signal detection.**
(DOCX)

## Author contributions

**Conceptualization:** Nonger Shen, Fangfang Ma.

**Data curation:** Nonger Shen, Qingxia Fang, Yue Wu, Lan Lan.

**Formal analysis:** Nonger Shen, Yue Wu.

**Methodology:** Nonger Shen, Qingxia Fang, Fangfang Ma.

**Software:** Nonger Shen.

**Writing – original draft:** Nonger Shen.

**Writing – review & editing:** Qingxia Fang, Fangfang Ma.

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
