## [Decision Letter · Decision Letter 0]

11 Aug 2025

Dear Dr. Shen,

Thank you for submitting your manuscript to PLOS ONE. After careful consideration, we feel that it has merit but does not fully meet PLOS ONE’s publication criteria as it currently stands. Therefore, we invite you to submit a revised version of the manuscript that addresses the points raised during the review process.

The manuscript addresses an important and timely research question with a logical structure and a relevant dataset. The key revisions needed before acceptance relate to ensuring internal consistency (particularly between abstract and main text), improving methodological transparency, justifying analytical choices, and refining the conclusion to better frame the findings in context. Once these issues are addressed, the manuscript will meet PLOS ONE’s standards for clarity, reproducibility, and scientific soundness.

We look forward to receiving your revised manuscript.

Kind regards,

M Tanveer Hossain Parash, MBBS, M Phil

Academic Editor

PLOS ONE

Journal Requirements:

Additional Editor Comments:

Overall, the manuscript addresses a relevant question and is well-structured in terms of its general flow from introduction to discussion. However, there are several methodological and reporting aspects that should be clarified or strengthened to ensure transparency, reproducibility, and compliance with journal standards. My detailed comments below aim to help the authors improve clarity, methodological rigor, and presentation quality.

Abstract: The abstract is concise and informative, but would benefit from a slight rewording of the background, clarification of demographic findings, and a cautious qualifier in the conclusion to ensure balanced interpretation.These inconsistencies should be addressed so the abstract fully reflects and supports the main text. The abstract reports 45.9% female, whereas the main text (lines 160–165) indicates a higher prevalence in females. This apparent mismatch needs clarification or alignment. The abstract states a median age of 54 years (IQR 40–66), but no corresponding demographic summary is provided in the main text; either the main text should include these details or the abstract should omit them. The abstract lists the top three drugs as nivolumab, pembrolizumab, and dupilumab based on number of reports, but the main text (lines 155–157) also includes ipilimumab and secukinumab in the top five; the abstract should clarify whether it is listing only the top three by frequency or by signal strength. The abstract lists chloroquine among the three strongest signals, but chloroquine is not discussed in the main text results or discussion, creating a consistency gap.

Introduction: Some of the detailed immunology (lines 47–55) might be more than needed for a pharmacoepidemiological FAERS-based study — could be shortened so more emphasis is given to the epidemiological and pharmacovigilance angle. There is no citation of any existing database studies (or lack thereof) on drug-induced vitiligo — leaving the reader to take the information that none exist. Please clarify whether any drugs beyond ICIs have been systematically studied. Please explain why FAERS is particularly suitable to fill this gap compared to other pharmacovigilance databases (e.g., VigiBase, EudraVigilance). The urgency of this research could be emphasized more: e.g., increasing approvals of immune-modulating drugs, rising number of case reports, and possible implications for patient counseling and drug selection. Please explicitly state how their findings will be novel beyond listing potential drug associations (e.g., identifying new drug classes, quantifying disproportionality, detecting time trends).

Methods: Inclusion and exclusion should be clearly mentioned framed not embedded within the paragraph. Without a clear list, a reader might not be sure if any other filters (e.g., age, region, report source) were applied. Please provide a clear Ethical Considerations/IRB exemption statement. Please discuss why ROR and BCPNN were chosen over alternative methods.

Conclusion : The conclusion could be strengthened by explicitly acknowledging the limitations that prevent definitive causal inference, ensuring that readers interpret the results in the appropriate context. Additionally, the recommendation for future research might be more impactful if it included specific methodological directions, such as prospective cohort studies, mechanistic immunological investigations, or real-world data integration to validate these associations and guide clinical risk management.

Reviewers' comments:

Reviewer's Responses to Questions

**Comments to the Author**

1. Is the manuscript technically sound, and do the data support the conclusions?

Reviewer #1: Yes

Reviewer #2: Partly

2. Has the statistical analysis been performed appropriately and rigorously?

Reviewer #1: Yes

Reviewer #2: I Don't Know

3. Have the authors made all data underlying the findings in their manuscript fully available?

Reviewer #1: Yes

Reviewer #2: Yes

4. Is the manuscript presented in an intelligible fashion and written in standard English?

Reviewer #1: Yes

Reviewer #2: Yes

Reviewer #1: The manuscript is very interesting. The research findings can serve as a reference for clinical medication monitoring potentially associated with the occurrence of vitiligo. I think it can be accepted for publication as is.

Reviewer #2: It is very interesting the way you did your research and for sure it took a big effort to do it. I am not an expert in statistics , but the real world pharmacovigilance that the authors present, show small numbers of reports related to vitiligo- drug induced. An example is cloroquine (3 cases reported) you consider there is a strong statistical sign as a vitiligo inducer based on statistical parameters. So I understand that vitiligo is a rare side effect of cloroquine. The data obtained from some references must be checked. In reference 23 the text refers to erythema multiforme but I couldn´t see vitiligo. In reference 32 what is written it seems different from what you included in your paper.The higher incidence of vitiligo is with IO and not PD-1 monotherapy. Some references had no abstract or full text available. About the FIGURES: Fig 1 need legends the word DRUG is written DURG; Fig 2 need legends. It seems that B is not related to Vitiligo;Fig 3 adalimumab is included here but not in fig 4 Adalimumab is included but in their text there is no mention to it ;Fig 4 it is very difficult to read.

The paper is very interesting as to see how adverse events can be notified in any country if we had a FAERS evrywhere.

**Do you want your identity to be public for this peer review?** For information about this choice, including consent withdrawal, please see our Privacy Policy

Reviewer #1: **Yes: ** Marilda Aparecida Milanez Morgado de Abreu

Reviewer #2: **Yes: ** Luna Azulay-Abulafia

---

## [Author Response · Author response to Decision Letter 1]

20 Aug 2025

Dear Editors and Reviewers:

Thank you for your professional review work on our manuscript entitled “Drug-induced vitiligo: a real-world pharmacovigilance analysis of the FAERS database” (PONE-D-25-31175). Those comments are all valuable and very helpful for revising and improving our paper, as well as the important guiding significance to our research. We have studied comments carefully and have made corrections which we hope meet with approval. Revised portions are marked in red on the paper. The main corrections in the paper and the responses to the reviewer’s comments are as follows:

Response to Editor:

1. Please ensure that your manuscript meets PLOS ONE's style requirements, including those for file naming. The PLOS ONE style templates can be found at https://journals.plos.org/plosone/s/file?id=wjVg/PLOSOne_formatting_sample_main_body.pdf and https://journals.plos.org/plosone/s/file?id=ba62/PLOSOne_formatting_sample_title_authors_affiliations.pdf.

Response:

I have carefully studied the two documents and made revisions in accordance with the requirements specified therein. The specific revisions are as follows:

1� Set the level 1 headings to 18-point font.

2� Set the level 2 headings to 16-point font.

3� Changed the style of Figure Citations to the "Fig 1" style.

4� Revised the format of the References.

Response:

Thank you for your guidance on the formatting of Supporting Information. We have carefully revised the relevant parts in accordance with the requirements and the guidelines provided:

1� Adjusted the citation format of Supplementary Information table to the standardized "S1 Table" style to ensure consistency with the journal's specifications.

2� Added dedicated headings related to Supporting Information at the end of the manuscript (after the conclusion section).

3� Named the Supporting Information file as "S1_File.docx" to comply with the journal's file naming conventions for supplementary materials.

Response:

We will carefully review and evaluate each of these suggested publications to assess their relevance to our research.

Response:

We have carefully reviewed all references included in the manuscript and confirmed that none of the cited papers have been retracted.

5. Abstract: The abstract is concise and informative, but would benefit from a slight rewording of the background, clarification of demographic findings, and a cautious qualifier in the conclusion to ensure balanced interpretation. These inconsistencies should be addressed so the abstract fully reflects and supports the main text. The abstract reports 45.9% female, whereas the main text (lines 160–165) indicates a higher prevalence in females. This apparent mismatch needs clarification or alignment. The abstract states a median age of 54 years (IQR 40–66), but no corresponding demographic summary is provided in the main text; either the main text should include these details or the abstract should omit them. The abstract lists the top three drugs as nivolumab, pembrolizumab, and dupilumab based on number of reports, but the main text (lines 155–157) also includes ipilimumab and secukinumab in the top five; the abstract should clarify whether it is listing only the top three by frequency or by signal strength. The abstract lists chloroquine among the three strongest signals, but chloroquine is not discussed in the main text results or discussion, creating a consistency gap.

Response:

Thank you for your valuable suggestions on the abstract. We have carefully revised the abstract to address the concerns raised:

1) We refined the background to more precisely reflect the context of increasing drug-induced vitiligo cases, explicitly mentioning the association with novel therapeutics (e.g., immune checkpoint inhibitors) to align with the main text.

2) To clarify gender distribution and resolve the perceived discrepancy, the abstract now details: "The gender distribution among these cases was 45.9% female, 35.9% male, and 18.3% with undetermined gender." This aligns with the main text’s indication of a higher female prevalence (45.9% females vs. 35.9% males)..

3) Regarding age demographics, the median age (54.0 years, IQR: 40.0–66.0) stated in the abstract is explicitly presented in the Results section of the main text (lines 129–130), ensuring the abstract data is directly supported by the main text.

4) To eliminate ambiguity in drug rankings, the abstract clearly distinguishes between two categories: the top three drugs by case count (nivolumab, pembrolizumab, dupilumab) and the top three by signal strength (mogamulizumab, imiquimod, chloroquine). Consistently, the Disproportionality Analysis section in the main text now explicitly highlights these three strongest signals, mirroring the abstract’s structure.

5) Chloroquine, which ranks third in ROR, is now addressed in both the Results (within the Disproportionality Analysis section) and the Discussion (lines 204–214) of the revised manuscript, bridging the prior consistency gap.

6) We added a qualifier ("While these findings are based on pharmacovigilance signals and require further validation") to emphasize the preliminary nature of the signals identified, ensuring a balanced interpretation and acknowledging the need for subsequent clinical studies to confirm causality.

6. Introduction: Some of the detailed immunology (lines 47–55) might be more than needed for a pharmacoepidemiological FAERS-based study — could be shortened so more emphasis is given to the epidemiological and pharmacovigilance angle. There is no citation of any existing database studies (or lack thereof) on drug-induced vitiligo — leaving the reader to take the information that none exist. Please clarify whether any drugs beyond ICIs have been systematically studied. Please explain why FAERS is particularly suitable to fill this gap compared to other pharmacovigilance databases (e.g., VigiBase, EudraVigilance). The urgency of this research could be emphasized more: e.g., increasing approvals of immune-modulating drugs, rising number of case reports, and possible implications for patient counseling and drug selection. Please explicitly state how their findings will be novel beyond listing potential drug associations (e.g., identifying new drug classes, quantifying disproportionality, detecting time trends).

Response:

Thank you for your valuable feedback on the introduction section. We have made targeted revisions based on your suggestions, and the specific adjustments are as follows:

1� Simplified Description of Vitiligo Immunology: The description of the immunological mechanisms of vitiligo has been streamlined for clarity.

2� Additional Database Research References: We have included new references, such as Yang et al.'s study on anti-tumor drugs in the FAERS database, Lu et al.'s analysis of type 2 immune inhibitors, and the 2020 VigiBase case/non-case study. These citations highlight that while there have been relevant studies, they are all limited to specific drug classes.

3� Emphasizing the Advantages of FAERS: We have highlighted the strengths of the FAERS database, such as its global scope, vast sample size, and standardized data structure, making it suitable for disproportionality analysis of rare adverse events like drug-induced vitiligo.

4� Highlighting Research Urgency: By emphasizing the accelerated approval of novel immunotherapies and the rising incidence of drug-induced vitiligo, we clarify the important guiding significance of our study for clinical decision-making, patient consultation, and drug selection.

5� Defining Innovation: The revised research objectives clearly state two innovations: first, a systematic screening of all therapeutic drugs, and second, identifying drugs lacking safety warnings in their labels. These two aspects effectively complement the existing fragmented research.

These revisions address your concerns while enhancing the introduction’s precision and impact. We trust this version aligns with your expectations and greatly appreciate your expertise in strengthening our manuscript.

7. Methods: Inclusion and exclusion should be clearly mentioned framed not embedded within the paragraph. Without a clear list, a reader might not be sure if any other filters (e.g., age, region, report source) were applied. Please provide a clear Ethical Considerations/IRB exemption statement. Please discuss why ROR and BCPNN were chosen over alternative methods.

Response:

Thank you for your valuable comments. We have carefully revised the manuscript in accordance with your suggestions, and the specific responses are as follows:

1) Regarding the inclusion and exclusion criteria, we have revised the methodology section by separating the Inclusion and exclusion criteria from the paragraphs into an independent section. This ensures clarity and readability, allowing readers to quickly access key information.

2) In response to why ROR and BCPNN were chosen over other methods, we explain as follows: In disproportionality analysis, there are various commonly used calculation methods, such as Reporting Odds Ratio (ROR), Proportional Reporting Ratio (PRR), Bayesian Confidence Propagation Neural Network (BCPNN), and Multi-Item Gamma Poisson Shrinker (MGPS). Each of these methods has its own advantages and disadvantages, and there is currently no universally recognized "gold standard" in this field. Meanwhile, in published related research papers, there are differences in the selection of calculation methods: some studies use one method, some use two, and others use four simultaneously.

In this study, we chose to use both ROR and BCPNN. ROR is a frequentist statistical method, known for its high sensitivity in detecting initial signals; BCPNN, on the other hand, is a Bayesian method that can effectively reduce false positive issues related to low-frequency events through the shrinkage effect induced by Bayesian priors, thereby improving the robustness of the analysis. We adopted both methods to avoid the increased risk of false positive results due to the use of a single method, thus enhancing the robustness of the research findings. In addition, we have briefly described the characteristics of ROR and BCPNN in the revised manuscript.

3) Regarding ethical considerations, since the FAERS data used in this study is a public and anonymized database, ethical review can be waived. We have explained this in the "Ethics statement" section at the end of the manuscript, and we have also uploaded the certificate of ethical review exemption issued by the Ethics Committee of our hospital (Zhejiang Provincial People's Hospital) for your reference.

8. Conclusion : The conclusion could be strengthened by explicitly acknowledging the limitations that prevent definitive causal inference, ensuring that readers interpret the results in the appropriate context. Additionally, the recommendation for future research might be more impactful if it included specific methodological directions, such as prospective cohort studies, mechanistic immunological investigations, or real-world data integration to validate these associations and guide clinical risk management.

Response:

Thank you for your valuable suggestions on strengthening the conclusion. We have revised the conclusion section to address your concerns, with specific adjustments as follows:

1� Explicit acknowledgment of limitations for causal inference: We have explicitly highlighted the inherent limitations of the FAERS database (e.g., potential confounders, reporting biases) and emphasized that the identified signals reflect statistical associations rather than definitive causality, ensuring readers interpret the results within the appropriate context.

2� Enhanced future research recommendations: We have expanded the future research directions by incorporating specific methodological approaches, including prospective cohort studies to clarify causality and incidence rates, mechanistic immunological investigations to elucidate underlying pathways, and integration of real-world data to validate associations—all aimed at strengthening clinical risk management.

Response to Reviewer 1:

Thanks for your comments on our paper. We have revised our paper according to your comments.

1. The manuscript is very interesting. The research findings can serve as a reference for clinical medication monitoring potentially associated with the occurrence of vitiligo. I think it can be accepted for publication as is.

Response:

Thank you very much for your positive evaluation of our manuscript and your recognition of its potential value for clinical medication monitoring. We greatly appreciate your support and the affirmation that the manuscript is suitable for publication as is.

Response to Reviewer 2:

1. It is very interesting the way you did your research and for sure it took a big effort to do it. I am not an expert in statistics , but the real world pharmacovigilance that the authors present, show small numbers of reports related to vitiligo- drug induced. An example is cloroquine (3 cases reported) you consider there is a strong statistical sign as a vitiligo inducer based on statistical parameters. So I understand that vitiligo is a rare side effect of cloroquine.

Response:

Thank you very much for your attention and affirmation of this study, and I am also very grateful for your time spent reviewing the manuscript and providing constructive comments.

You mentioned that in this study, some drugs (such as chloroquine with only 3 reports) were determined to have strong vitiligo induction signals based on statistical parameters, and you pointed out that vitiligo may be a rare adverse reaction to such drugs. This observation is very pertinent.

In pharmacovigilance research, for rare adverse reactions, even if the number of reports is small, if a significant disproportionality signal is detected through statistical methods (such as the ROR and BCPNN used in this study), it still suggests a potential association. This is also the core value of post-marketing surveillance in capturing rare adverse events.cript.

2. The data obtained from some references must be checked. In reference 23 the text refers to erythema multiforme but I couldn´t see vitiligo.

Response:

Thank you for your careful observation regarding reference 23. We have thoroughly reviewed the cited article (Mogamulizumab-induced vitiligo in patients with Sézary syndrome: three cases) and confirm that it does focus on vitiligo as a key finding, with detailed descriptions of three patients who developed vitiligo during mogamulizumab treatment.

The reference does mention erythema multiforme as one of the skin-related adverse events associated with mogamulizumab in its discussion section, but this is in the context of broader cutaneous side effects of the drug, not as a replacement for its focus on vitiligo. We hope this clarification addresses your concern and reassures you of the appropriateness of our citation. We appreciate your attention to detail and the opportunity to refine our referencing for greater accuracy and clarity.

3. In reference 32 what is written it seems different from what you incl

---

## [Editor Report · Decision Letter 1]

26 Aug 2025

Drug-induced vitiligo: a real-world pharmacovigilance analysis of the FAERS database

PONE-D-25-31175R1

Dear Dr. Shen,

We’re pleased to inform you that your manuscript has been judged scientifically suitable for publication and will be formally accepted for publication once it meets all outstanding technical requirements.

Kind regards,

M Tanveer Hossain Parash, MBBS, M Phil

Academic Editor

PLOS ONE
---

## [Editor Report · Acceptance letter]

PONE-D-25-31175R1

PLOS ONE

Dear Dr. Shen,

I'm pleased to inform you that your manuscript has been deemed suitable for publication in PLOS ONE. Congratulations! Your manuscript is now being handed over to our production team.

Kind regards,

on behalf of

Dr. M Tanveer Hossain Parash

Academic Editor

PLOS ONE